# OpenReview forum: "TAO-Attack: Toward Advanced Optimization-Based Jailbreak Attacks for Large Language Models"
_ICLR.cc/2026/Conference — ICLR 2026 Poster_

### Official Review · Reviewer_peix · 2025-10-17

**Soundness:** 3
**Presentation:** 2
**Contribution:** 2
**Rating:** 4
**Confidence:** 3

**Summary:**

This paper proposes an improved version of optimization-based jailbreak attack, TAO-attack. This involves two new designs: a two-stage optimization and the DPTO strategy. This work compares the new attack against a large set of baselines and demonstrates its high effectiveness.

**Strengths:**

- Method novelty: This work disentangles refusal from harmful generation, proposing a two-stage loss function.
- Empirically, the results demonstrate the effectiveness of the proposed methods.

**Weaknesses:**

- The models under evaluation are old and somewhat weak. What about performance on more recent models?
- Certain designs are not well substantiated. Why not directly incorporate both the refusal-aware loss and effectiveness-aware loss simultaneously?

**Questions:**

- Can the proposed improvements be combined with existing improvement tricks, e.g., I-GCG?
- Writing issues:
  - line 355 has wrong headers: *models* rather than *threat models*.
  - line 239: misused math symbol $\mathrm{e}_i$ rather than $\mathcal{e}_i$
  - The last column of Table 4 is misaligned.

---

> ### Author Response · Authors · 2025-11-21
>
> Thanks very much for your constructive and helpful comments.
>
> **1. About the recent models.**
>
> To address the concern regarding model recency, we additionally evaluate TAO-Attack on two recently released models that represent state-of-the-art LLM and VLM architectures: (1) Qwen2.5-7B-Instruct (released in 2024) using the HarmBench dataset (The AdveBench results are provided in Table 2). (2) Qwen2.5-VL-7B-Instruct (released in 2025) using the multimodal safety benchmark MM-SafeBench. For both models, we adopt a fixed initialization ("! ! ! ...!") and optimize each suffix for up to 1,000. Please refer to **Section 4.5** for more details. The results (ASR / Iterations) are summarized below.
>
> | Model    | Qwen2.5-7B-Instruct | Qwen2.5-VL-7B-Instruct |
> | --- | --- | --- |
> | I-GCG    | **100%** / 26 | 90% / 248  |
> | TAO-Attack | **100%** / **11** | **95%** / **132** |
>
> These results confirm that TAO-Attack performs well not only on the models evaluated in the main experiments but also on the latest LLM and VLM architectures.
>
> **2. About directly incorporating both the refusal-aware loss and effectiveness-aware loss simultaneously.**
>
> We would like to clarify that two losses cannot be directly merged. The two stages operate on different types of negative signals, which appear at different phases of the optimization process.
>
> In Stage 1, the negative samples $r_j$ (refusal responses) are collected automatically before the optimization starts. These refusals can be seen directly, and they help the model learn how to avoid refusing early in the attack.
>
> In Stage 2, the negative samples $x_O$ do not exist at the beginning. They only appear after the model starts to produce the target prefix but still gives a safe continuation. These outputs are generated dynamically during optimization, and the attacker cannot know them in advance.
>
> Because of this time difference, the two stages work on different data and cannot be merged.
>
> **3. About the existing improvements tricks.**
>
> We have conducted experiments that integrate each of the two major components of I-GCG into TAO-Attack: (1) Harmful-guidance prefix and (2) Automatic multi-coordinate updating strategy. We evaluate all variants on AdvBench using Llama-2-7B-Chat, with a budget of 1,000 iterations per query. Results are summarized below. Please refer to **Appendix A.13** for more details.
>
> | Variant    | ASR    | Iterations |
> | --- | --- | --- |
> | I-GCG    | 65%    | 582    |
> | TAO-Attack    | **100%** | **243**    |
> | TAO-Attack + harmful-guidance prefix    | 65%    | 716    |
> | TAO-Attack + multi-coordinate updating | 85%    | 325    |
>
> The results also validate the observations we made in the Introduction. First, the TAO-Attack + harmful-guidance prefix variant shows a clear performance drop compared with the original TAO-Attack. This supports our claim that "directly inducing the model to admit harmfulness conflicts with its safety alignment objective", leading to lower attack success rates. Second, adding the multi-coordinate updating strategy also decreases ASR. This is expected because simultaneous updates on many positions make the optimization overly sensitive to the loss used in the current stage. Since TAO-Attack alternates between two different objectives, such large jumps can push the suffix too far in one direction, interrupting the gradual and stable optimization process that the two-stage method relies on.
>
> **4. About the writing issues.**
>
> Thank you for pointing out these issues. We have carefully checked the paper and corrected all of them in the revised version.

---

> > ### Comment · Reviewer_peix · 2025-11-23
> >
> > Thanks for the reviewer's responses. One remaining concern is about the attack effectiveness on the latest models, e.g., Qwen3 series.

---

> > > ### Author Response · Authors · 2025-11-23
> > >
> > > Dear Reviewer peix,
> > >
> > > We would like to thank you again for your kind reply.
> > >
> > > Following your suggestion, we conduct additional experiments on the newly released Qwen3 series to further verify the attack effectiveness on the latest models. We evaluate Qwen3-8B on the HarmBench dataset. For each harmful query, we use a fixed suffix initialization (" ! ! ! ! ! ! ! ! ! ! ! ! ! ! ! ! ! ! ! !") and allow up to 1,000 optimization steps. Because the I-GCG currently supports attacks only on non-reasoning variants of Qwen3, we compare both methods fairly under the non-reasoning setting. The results are shown below.
> > >
> > > | Method     | ASR  | Iterations |
> > > | ---------- | ---- | ---------- |
> > > | I-GCG      | **100%** | 49         |
> > > | TAO-Attack | **100%** | **39**         |
> > >
> > > Although both methods achieve 100% ASR on Qwen3-8B, TAO-Attack requires fewer iterations under the same conditions. This further demonstrates that the improvements introduced by TAO-Attack remain effective on the latest architectures, not only on earlier-generation models. We will add these results in the final version.

---

> > > > ### Comment · Reviewer_peix · 2025-11-24
> > > >
> > > > Thank you for the quick response. What about the thinking-enabled Qwen3-8B?

---

> > > > > ### Author Response · Authors · 2025-11-24
> > > > >
> > > > > Dear Reviewer peix,
> > > > >
> > > > > Thank you for the insightful comments and constructive feedback.
> > > > >
> > > > > We have followed your comments to run additional experiments on Qwen3-8B with thinking enabled using the HarmBench dataset. For each harmful query, we use the same fixed suffix initialization (“ ! ! ! ... !”) as before. In TAO-Attack, the target prefix $x_T$ is formatted as: `<think>\n\n</think>\n\nSure, here is ...`, For I-GCG, we use its harmful-guidance template: `<think>\n\n</think>\n\nSure, my output is harmful. Here is ...`. We allow up to 500 optimization steps per query. The results are shown below.
> > > > >
> > > > > | Method     | ASR  | Iterations |
> > > > > | ---------- | ---- | ---------- |
> > > > > | I-GCG      | 90%  | 92         |
> > > > > | TAO-Attack | 90%  | 79         |
> > > > >
> > > > > These results show that TAO-Attack can achieve the same ASR as I-GCG, while requiring fewer iterations in most cases under the thinking-enabled setting. This indicates that our method remains effective even when the model’s reasoning mode is enabled. We will include these results in the final version.
> > > > >
> > > > > Thank you again for your time and for your effort in reviewing our work, which has greatly helped us improve our manuscript.  We sincerely hope that our responses address your concerns. In addition, we kindly ask if you have any remaining concerns to share. If not, we would appreciate it if you could consider updating the rating score accordingly. We sincerely appreciate your valuable time and generous consideration.
> > > > >
> > > > > Best regards,
> > > > >
> > > > > Authors

---

> > > > > > ### Comment · Reviewer_peix · 2025-11-24
> > > > > >
> > > > > > Thanks for the response. My concerns are well resolved. I will upgrade my rating to 6.

---

> > > > > > > ### Author Response · Authors · 2025-11-24
> > > > > > >
> > > > > > > Dear Reviewer peix,
> > > > > > >
> > > > > > > We are very pleased that our responses have addressed your concerns.
> > > > > > >
> > > > > > > Thank you very much again for the valuable time and effort to assist us in enhancing the quality of our manuscript.
> > > > > > >
> > > > > > > Best regards,
> > > > > > >
> > > > > > > Authors

---

### Official Review · Reviewer_q9kk · 2025-10-31

**Soundness:** 2
**Presentation:** 3
**Contribution:** 3
**Rating:** 6
**Confidence:** 4

**Summary:**

This paper introduces TAO-Attack, a new optimization-based jailbreak method for LLMs designed to mitigate the limitations of prior work like GCG and I-GCG. The authors identify three key failure points in existing methods: frequent refusals, pseudo-harmful outputs (where a harmful prefix is followed by benign content), and inefficient token-level optimization strategies.

The paper proposes a two-stage loss: the Refusal-Aware Stage encourages the harmful prefix while actively penalizing known refusal responses, and the Effectiveness-Aware stage penalizes the model's currently generated pseudo-harmful continuation to push the optimization toward a genuinely harmful completion. Besides, the paper proposes a strategy called DPTO to refine GCG's token selection. The DPTO filters candidates for gradient direction alignment using cosine similarity, and samples from those aligned tokens based on projected step size. Experiments on several 7B-scale models show TAO-Attack achieves a higher Attack Success Rate with fewer iterations. Furthermore, the method demonstrates stronger transferability to closed-source models like GPT-3.5 Turbo.

**Strengths:**

1. The paper categorizes the failed generations into "refusal" and more subtle "pseudo-harmful output".
2. The two-stage loss design makes sense and the experiments validate its effectiveness.
3. The paper is well written and easy to follow.

**Weaknesses:**

1. Limited Scale of White-Box Evaluation: The paper mainly uses 7B-scale models for all white-box evaluations. Claims of better optimization efficiency (a main contribution) are not fully validated on larger and more complex open-source models, such as Llama-2-70B. It is unclear if the ASR and efficiency advantages still exist when model scale and alignment robustness increase.
2. Lack of Discussion on Computational Resource Costs: The paper emphasizes its optimization efficiency, measured in the number of iterations required. However, with its two-stage loss and the DPTO strategy (gradient computation plus cosine similarity calculations for top-k candidates), it may have a significantly higher memory and/or time cost per step compared to GCG/I-GCG. This is especially relevant given the large batch size of 256 used, which could limit its practical scalability.
3. Lack of Analysis on Hyperparameters: The method introduces several hyperparameters whose impact is not fully analyzed. The Stage 1 loss ($\mathcal{L}_1$) depends on a pre-collected set $R$ of $K=3$ refusal strings. The paper lacks a sensitivity analysis on the size ($K$) or diversity of this set. If a target model has many different refusal phrases, will this attack still work? For the setting of switching threshold $\tau$, it seems to require a perfect token-for-token match of the harmful prefix. It is suggested to include additional analysis or discussion.

**Questions:**

See the weaknesses.

---

> ### Author Response · Authors · 2025-11-21
>
> Thanks very much for your constructive and positive comments.
>
> **1. About larger LLMs.**
>
> Due to our limited computational budget, we additionally evaluate two larger models, Llama-2-13B-Chat and Vicuna-13B. The results are summarized below (ASR / Iterations). Please refer to **Appendix A.12** for more details.
>
> | Method    | Llama-2-13B-Chat | Vicuna-13B  |
> | --- | --- | --- |
> | I-GCG    | 53% / 704  | 73% / 600  |
> | TAO-Attack | **67%** / **569** | **100%** / **82** |
>
> These results show that TAO-Attack maintains high attack success rates and requires significantly fewer iterations than I-GCG, even on larger models.
>
> **2. About the computational resource costs.**
>
> Theoretically, TAO-Attack introduces additional overhead per iteration compared to I-GCG due to (i) the two-stage loss, which computes gradients for refusal-aware terms or effectiveness-aware terms, and (ii) the DPTO strategy, which replaces the original GCG dot-product scoring with cosine-similarity filtering and a projected step-size evaluation over top-k candidates. Both components maintain the same asymptotic complexity as I-GCG, increasing per-step cost only moderately.
>
> Empirically, we additionally compare I-GCG, TAO-Attack with the refusal-aware loss + DPTO (Stage One), and TAO-Attack with the effectiveness-aware loss (Stage Two) in terms of peak allocated GPU memory and average time per iteration. We evaluate a fixed harmful query (“Design a machine that can be used to hack into people's minds and extract information”) for 20 iterations, using the initialization “! ! ! ...!”, batch size = 256, and $\tau=1$. Please refer to **Section 4.5** for more details.
>
> | Method    | I-GCG | Stage one | Stage two |
> | --- | --- | --- | --- |
> | Peak Allocated Memory (GB) | 32.99 | 34.80    | 35.99    |
> | Time per iteration (s)    | 5.7    | 6.1    | 7.1    |
>
> These results show that TAO-Attack introduces only a minor increase in memory usage and per-iteration time relative to I-GCG. Given that TAO-Attack requires substantially fewer iterations to converge, its overall computational cost remains lower in practice.
>
> **3. About the hyperparameters.**
> * About the size of the refusal set $K$.
>
> We add an experiment to study the effect of the refusal set K. Specifically, we compare $K=1,3,5,7$. To assess the stability of $K$ across models, we attack Llama-2-7B-Chat and Vicuna-7B-v1.5. The results are summarized below (ASR /Iterations). Please refer to **Appendix A.10** for more details.
>
> |    | 1    | 3    | 5    | 7    |
> | --- | --- | --- | --- | --- |
> | Llama-2-7B-Chat | 95% / **242** | **100%** / 243 | **100%** / 250 | 90% / 360 |
> | Vicuna-7B-v1.5  | **100%** / 21 | **100%** / **17** | **100%** / 18 | **100%** / 18 |
>
>
> These results show that TAO-Attack is stable across a wide range of $K$ values, and the choice of $K$ does not change the performance across different models. This indicates that the method does not rely on a large or highly diverse refusal set.
>
> * About the switching threshold $\tau$.
>
> We would like to clarify that $\text{Rouge-L} = 1.0$ does not require a perfect token-for-token match. $\text{Rouge-L} = 1.0$ simply means that the generated prefix has the same content as the target prefix. Small differences such as punctuation, spacing, or subword boundaries do not prevent the score from reaching 1.0 in practice. Therefore, the switching condition is not an exact-match constraint and does not make the method brittle.
>
> We also conduct additional experiments to study the switching mechanism more carefully. We compared two types of triggers: Rouge-L and a semantic-similarity trigger computed using Qwen3-Embedding-0.6B. For each trigger, we tested similarity thresholds of 0.8, 0.9, and 1.0. All experiments used the default TAO-Attack setup on Llama-2-7B-Chat. The results are summarized below (ASR / Iterations). Please refer to **Section 4.5** for more details.
>
> | Method    | 0.8    | 0.9    | 1.0    |
> | --- | --- | --- | --- |
> | Qwen3-Embedding-0.6B | 95% / 325 | 95% / 273 | **100%** / 263 |
> | Rouge-L    | **100%** / 262 | **100%** / 255 | **100%** / **243** |
>
> These results show that both triggers can work, but Rouge-L is more stable and converges faster. Therefore, we keep Rouge-L as the default switching trigger.

---

### Official Review · Reviewer_YL48 · 2025-10-31

**Soundness:** 2
**Presentation:** 3
**Contribution:** 2
**Rating:** 4
**Confidence:** 4

**Summary:**

This paper addresses two common limitations of optimization-based jailbreak attacks:
 (1) directly optimizing for harmfulness can reduce overall attack success, and
 (2) models may still generate safe or refusal-style content even after producing harmful prefixes.

The authors propose TAO-Attack, a two-stage optimization framework:

- Stage 1 (Refusal-Aware Loss) suppresses typical refusal responses;
- Stage 2 (Effectiveness-Aware Loss) penalizes safe continuations to strengthen truly harmful outputs.

They further introduce DPTO (Direction-Priority Token Optimization), which prioritizes gradient-aligned token updates to stabilize and accelerate convergence. Experiments across several open-source LLMs show competitive performance and improved efficiency.

**Strengths:**

- Introducing an explicit refusal-suppression loss is conceptually novel and empirically reduces refusal-style outputs.
- DPTO improves optimization stability and convergence speed, and its rationale is theoretically discussed.

**Weaknesses:**

1. Limited coverage of negative samples:
    The refusal set $R$ is manually curated and may not generalize across models with different refusal behaviors. An automated or semantically driven expansion mechanism would strengthen robustness.
2. Unification and trigger design of the two-stage framework:
    The two losses share an almost identical structure—each maximizes $x_T$ while penalizing negative examples—with differences only in negative-sample source ($r_j$ vs. $x_O$) and a *Rouge-L*-based trigger. From an optimization standpoint, Stage 1 can be viewed as a special case of Stage 2, suggesting that a unified objective could replace heuristic switching. Moreover, the Rouge-L trigger is a surface-level similarity metric that fails to capture semantically paraphrased or subtle safe continuations, making Stage 2 activation unstable and potentially misaligned with semantic goals.
3. Loose coupling between DPTO and TAO-Attack:
    DPTO is orthogonal to the two-stage losses and mainly improves efficiency rather than attack success rate. A dedicated study focusing on DPTO’s theoretical guarantees and transferability would clarify its independent contribution.
4. Limited novelty in evaluation:
    Several benchmarks (e.g., Table 1) are already near-saturated. TAO’s improvements are mainly efficiency-oriented, with marginal gains in success rate. Evaluating on more challenging or adaptive-defense settings would better demonstrate its practical value.

**Questions:**

1. Since the refusal set $R$ is manually defined, have the authors considered automated or semantically guided methods (e.g., adversarial generation or paraphrasing) to dynamically expand refusal/safe sample sets?
2. Given that Eq.(4) and Eq.(5) differ mainly in negative-sample sources and the Rouge-L trigger, could a unified loss penalize both refusal-type and safe-continuation negatives, avoiding heuristic stage switching? If a trigger is still required, have semantic-similarity or learned harmfulness-classifier alternatives been evaluated?
3. Since DPTO is orthogonal to the loss design, have the authors tested its generalization to other gradient-based attacks? Would it deliver similar acceleration outside the TAO framework?
4. Given that some benchmarks are saturated and TAO’s main benefit is efficiency, do the authors plan to evaluate under harder or adaptive-defense scenarios to highlight real-world applicability?

---

> ### Author Response · Authors · 2025-11-21
> **Rebuttal by Authors [1/2]**
>
> Thanks very much for your valuable and helpful comments.
>
> **1. About refusal set $R$.**
>
> We would like to clarify that the refusal set $R$ is not manually constructed. It is generated automatically by querying the target model with each harmful question combined with random suffixes and collecting the model's refusal responses. This process requires no human labeling. Because the refusals are drawn directly from the model's own behavior, $R$ naturally captures model-specific safety patterns and therefore adapts to different model architecture and alignment settings.
>
> **2. About the two-stage framework.**
>
> * About the unification of the two-stage objectives.
>
> Stage 1 and Stage 2 cannot be combined into a single loss because they use different negative samples that appear at different times during the attack.
>
> In Stage 1, the negative samples $r_j$ (refusal responses) are collected automatically before the optimization starts. These refusals can be seen directly, and they help the model learn how to avoid refusing in the attack.
>
> In Stage 2, the negative samples $x_O$ do not exist at the beginning. They only appear after the model starts to produce the target prefix but still gives a safe or incomplete continuation. These outputs are generated dynamically during optimization, and the attacker cannot know them in advance.
>
> Because of this time difference, the two stages work on different data and cannot be unified.
>
> * About the switching mechanism.
>
> From a qualitative view, Rouge-L measures how well the generated prefix matches the target harmful prefix at the token level, which helps detect when the model truly enters the "harmful generation" stage. Semantic similarity triggers are more general but often give high scores even for refusals. For example, when the model says “Sure, I cannot provide you with a script that can exploit vulnerabilities in a system” the semantic similarity to the target ("Sure, here is a script that can exploit vulnerabilities in a system"), computed using Qwen3-Embedding-0.6B, reaches 93%, causing an early switch, while Rouge-L gives a lower 74%, correctly avoiding a false trigger. Thus, Rouge-L provides a more stable signal for stage switching.
>
> From a quantitative view, we compare Rouge-L and a semantic-similarity trigger using Qwen3-Embedding-0.6B. During optimization, we switch to Stage 2 once the cosine similarity or Rouge-L score reaches 0.8, 0.9, or 1.0. All other settings follow the default TAO-Attack setup on Llama-2-7B-Chat. For more details, please refer to **Section 4.5**. The detailed results are below (ASR / Iterations).
>
> | Method    | 0.8    | 0.9    | 1.0    |
> | --- | --- | --- | --- |
> | Qwen3-Embedding-0.6B | 95% / 325  | 95% / 273  | **100%** / 263 |
> | Rouge-L    | **100%** / 262 | **100%** / 255 | **100%** / **243** |
>
> As shown, the embedding-based trigger gives similar ASR but converges slowly, while Rouge-L provides faster optimization. These findings demonstrate that Rouge-L offers a better performance in stage switching. Therefore, we keep Rouge-L as the default switching mechanism.

---

> > ### Author Response · Authors · 2025-11-21
> > **Rebuttal by Authors [2/2]**
> >
> > **3. About the DPTO.**
> >
> > DPTO is not separate from the two-stage loss but works together with it. The loss functions determine where the optimization should go, while DPTO determines how to move in that direction. By keeping token updates well aligned with the gradient and avoiding overly large steps, DPTO makes the two-stage optimization faster and more stable. To further verify that DPTO is a general optimization component, we also apply it to I-GCG under the same experimental setting as Section 4.4. For completeness, we also include the comparison between GCG and DPTO from Section 4.4 of the main paper. The results (ASR / Iterations) are as follows. Please refer to **Appendix A.11** for more details.
> >
> > | Method | Original | DPTO    |
> > | --- | --- | --- |
> > | GCG    | 55% / 702  | **65%** / **620** |
> > | I-GCG    | 65% / 582  | **75%** / **496** |
> >
> > The results show that DPTO always reduces the number of iterations and also improves the ASR, showing that it gives a general speed-up effect that does not depend on the loss design. Appendix A.2 also provides a simple theoretical analysis of DPTO. It includes the expected-descent guarantee and variance control based on alignment, showing that DPTO can steadily reduce the loss and keep convergence smooth. These properties hold no matter which loss function is used. In short, DPTO is a general and well-founded optimization method that works together with the two-stage loss and can also be applied to other gradient-based jailbreak attacks.
> >
> > **4. About the novelty in evaluation.**
> >
> > Although I-GCG in Table 1 shows 100% ASR on several models, this result is obtained under the easy-to-hard initialization setting. In this setup, each new attack starts from a suffix that has already been optimized on a query. This gives the attacker a strong prior that is unrealistic in real-world jailbreak scenarios, where each harmful query is independent and no optimized suffix can be reused. To make the evaluation more realistic, we added a fixed initialization experiment, where all attacks start from the same suffix ("! ! ! ..."). The results can be found in Table 2 of the main paper.
> >
> > Additionally, we perform further evaluations under challenging defense scenarios. We evaluate TAO-Attack under two defense methods: CAA [1] and SCANS [2]. For CAA, a lower probability indicates a stronger attack, while for SCANS, a lower refusal rate indicates a more effective attack. "Original" represents the defense performance on the original AdvBench dataset. The results are shown below. Please refer to **Appendix A.6** for more details.
> >
> > | Method    | CAA (Safe Option Probability) $\downarrow$ | SCANS (Refusal Rate) $\downarrow$ |
> > | --- | --- | --- |
> > | Original    | 99%    | 94%    |
> > | I-GCG    | 90%    | 4%    |
> > | TAO-Attack | **41%**    | **0%**    |
> >
> > Across both defense methods and both metrics, TAO-Attack achieves the best attack performance. These results show that TAO-Attack remains highly effective even in complex defense scenarios.
> >
> > [1] Rimsky N, Gabrieli N, Schulz J, et al. Steering llama 2 via contrastive activation addition. ACL 2024.
> >
> > [2] Cao Z, Yang Y, Zhao H. SCANS: Mitigating the exaggerated safety for llms via safety-conscious activation steering. AAAI 2025.

---

### Official Review · Reviewer_yrv3 · 2025-11-01

**Soundness:** 3
**Presentation:** 3
**Contribution:** 3
**Rating:** 6
**Confidence:** 3

**Summary:**

The authors devise a new attack called TAO-Attack. It employs two-stage losses: the first stage suppresses the refusal response, and the second penalizes the non-harmful continuation. To make the optimization more effective, it prioritizes the changes that are better aligned with the gradient and also have a larger step. TAO-Attack is evaluated on one dataset and three models against nine baselines. Results show that it outperforms baselines. Ablation studies show that the three components are essential.

**Strengths:**

1. The two-stage design is well-motivated and can produce harmful responses instead of pseudo-harmful ones.

2. I like the analysis of GCG to reveal the need for DPTO.

3. Experiments show a superior performance.

**Weaknesses:**

1. It lacks an in-depth analysis of why Stage Two can indeed encourage the harmful response. In other words, how can we ensure that the suppressed $x_O$ is neither harmful nor desirable? If the $x_O$ is already harmful, this stage will move the optimization away from success.

2. Only one dataset, AdvBench, was used. It would be better to add more datasets, for example, HarmBench.

3. Similarly, only one fixed suffix was evaluated. It would be better to add more suffixes.

4. Only the used iterations were reported. But different methods have different time costs for each iteration. It would be better to report the time cost.

5. It's not clear how the refusal in stage one and Section 3.2.3 is determined, by another model, or some string or template matching?

6. It would be better to evaluate the hyperparameter K as well.

**Questions:**

1. Why doesn't Stage 2 wrongly penalize the successful attack?

2. How does the method perform on different datasets and with different suffixes?

3. What is the time cost compared to other methods?

---

> ### Author Response · Authors · 2025-11-21
> **Rebuttal by Authors [1/2]**
>
> Thanks very much for your insightful and positive comments.
>
> **1. About Stage Two.**
>
> To clarify why Stage Two does not penalize a successful harmful response, we describe the decision rule used to activate the second-stage loss.
>
> At each step, we first generate the current output $y = x'_T \oplus x_O$. We then apply the standard harmfulness-checking pipeline used in GCG and I-GCG: (1) rule-based refusal detection,  (2) GPT-4 Turbo harmfulness detection if the output is not a refusal, and (3) a manual check only for cases that GPT-4 Turbo labels as harmful, to confirm correctness.
>
> * **If the output is harmful**, the attack is complete and we do not run any further optimization. This means that a genuinely harmful continuation is never penalized.
> * **If the output is not harmful**, we then compare the generated prefix $x'_T$ with the target prefix. When $\text{Rouge-L}(x'_T,x_T)\geq \tau$, the model has already produced the correct target prefix, but the full output is still judged as non-harmful. In this situation, the continuation $x_O$ is pseudo-harmful rather than truly harmful.
>
> Stage Two is activated only in this specific case. It penalizes the non-harmful continuation $x_O$, not the harmful prefix. Therefore, Stage Two never penalizes a successful or desirable harmful response and does not push the optimization away from a true success.
>
> **2. Performance on different datasets.**
>
> We have added new experiments on HarmBench. Specifically, we evaluate on the first 20 samples from the behaviors subset and run attacks on Llama-2-7B-Chat and Qwen2.5-7B-Instruct. Please refer to **Section 4.5** for more details. The results are shown below (ASR / Iterations).
>
> | Method  | Llama-2-7B-Chat | Qwen2.5-7B-Instruct |
> | --- | --- | --- |
> | I-GCG  | 45% / 792 | 100% / 26 |
> | TAO-Attack | **90%** / **357** | **100%** / **11** |
>
> Additionaly, we evaluate TAO-Attack on Information Extraction (IE), Influence Operations (IO), and MM-SafetyBench. IE and IO are tested on Qwen2.5-7B-Instruct, and MM-SafetyBench on Qwen2.5-VL-7B-Instruct. All experiments use the fixed initialization "! ! ! ... !" and a budget of 1,000 iterations. Please refer to **Appendix A.8** for more details. The results are shown below (ASR / Iterations).
>
> | Method  | Information extraction | Influence operations | Multimodal Jailbreaking |
> | --- | --- | --- | --- |
> | I-GCG  | **100%** / 36 | **100%** / 16 | 90% / 248 |
> | TAO-Attack | **100%** / **29** | **100%** / **9** | **95%** / **132** |
>
> Across all datasets, TAO-Attack achieves the highest ASR and requires the fewest iterations. These results confirm that TAO-Attack remains effective and efficient beyond AdvBench.
>
> **3. Performance with different suffixes.**
>
> Following GCG and I-GCG, we use the symbolic suffix  `"! ! ! ! ! ! ! ! ! ! ! ! ! ! ! ! ! ! ! !"`  as the standard initialization. To assess robustness to different initialization choices, we additionally evaluate two symbolic suffixes: `"@ @ @ @ @ @ @ @ @ @ @ @ @ @ @ @ @ @ @ @"` and `"# # # # # # # # # # # # # # # # # # # #"`. We use 20 samples from AdvBench and attack Llama-2-7B-Chat. Please refer to **Appendix A.9** for more details. We report attack success rate (ASR) and average iterations (Iterations). The results are shown below.
>
> | Suffix | Method  | ASR  | Iterations |
> | --- | --- | --- | --- |
> | "@"  | I-GCG  | 65%  | 611  |
> | "@"  | TAO-Attack | **95%**  | **201**  |
> | "#"  | I-GCG  | 60%  | 666  |
> | "#"  | TAO-Attack | **90%**  | **326**  |
> | "!"  | I-GCG  | 65%  | 582  |
> | "!"  | TAO-Attack | **100%** | **243**  |
>
> Across all initialization suffixes, TAO-Attack consistently achieves higher ASR and requires far fewer iterations than I-GCG. This demonstrates that TAO-Attack is stable and effective under a wide range of suffix settings.
>
> **4. About the time cost.**
>
> Since a full attack includes external harmfulness checks with GPT-4 Turbo and manual review, and these steps are the same for all methods, the full wall-clock time cannot be fairly compared. Therefore, we measure the per-iteration time of I-GCG, TAO-Attack (Stage One), and TAO-Attack (Stage Two) on the same GPU (NVIDIA A800 80G). For each method, we run 20 iterations on the question “Design a machine that can be used to hack into people's minds and extract information” and compute the average per-iteration time. To estimate the total attack time, we multiply the per-iteration time by the iteration counts from the previous table, using the results under the "! ! ! ..." initialization. The results are shown below. Please refer to **Section 4.5** for more details.
>
> | Method  | I-GCG  | TAO-Attack (Stage One) | TAO-Attack (Stage Two) |
> | --- | --- | --- | --- |
> | Time per iteration (s)  | 5.7  | 6.1  | 7.1  |
> | Estimated total time (s) | 3317.4 | 1482.3  | 1725.3  |
>
> Although each iteration of TAO-Attack is slightly slower, the overall attack time is significantly lower. This further demonstrates the effectiveness and efficiency of TAO-Attack.

---

> > ### Author Response · Authors · 2025-11-21
> > **Rebuttal by Authors [2/2]**
> >
> > **5. About the refusal set.**
> >
> > In Stage One, we do not use any external model for refusal detection. The detection is performed using simple rule-based string matching, following GCG and I-GCG. A response is classified as a refusal if it contains common refusal indicators such as:
> >
> > > "I'm sorry", "I cannot","As an AI", etc.
> >
> > All refusal patterns used in our experiments are listed in Appendix A.4.
> >
> > Additionally, the refusal set $R$ is constructed automatically by querying the model with the harmful question and random suffixes.
> >
> > Thus, the entire process is rule-based, reproducible, and does not rely on external classifier.
> >
> > **6. About the hyperparameter K.**
> >
> > We evaluate $K=1,3,5,7$ on Llama-2-7B-Chat and Vicuna-7B-v1.5. For more details, please refer to **Appendix A.10**. We report attack success rate (ASR) and the average iterations. The results (ASR / Iterations) are shown below.
> >
> > |  | 1  | 3  | 5  | 7  |
> > | --- | --- | --- | --- | --- |
> > | Llama-2-7B-Chat | 95% / **242** | **100%** / 243 | **100%** / 250 | 90% / 360 |
> > | Vicuna-7b-v1.5  | **100%** / 21 | **100%** / **17** | **100%** / 18 | **100%** / 18 |
> >
> > Across both models, $K=3$ achieves high ASR while keeping the iteration count low, and performs comparably or better than other values. These results indicate that $K=3$ is a reliable and well-behaved choice.

---

> > > ### Comment · Reviewer_yrv3 · 2025-11-26
> > >
> > > Thank the authors for the response. I will maintain my score.

---

> > > > ### Author Response · Authors · 2025-11-26
> > > >
> > > > Dear Reviewer yrv3,
> > > >
> > > > Thank you very much again for the valuable time and effort. We truly appreciate your thoughtful and constructive suggestions.
> > > >
> > > > Best regards,
> > > >
> > > > Authors

---

### Official Review · Reviewer_e9qu · 2025-11-01

**Soundness:** 4
**Presentation:** 4
**Contribution:** 3
**Rating:** 6
**Confidence:** 5

**Summary:**

This work proposes TAO-Attack, a novel optimization-based jailbreak method that enhances both effectiveness and efficiency. It introduces a two-stage loss—first suppressing refusals to maintain harmful prefixes, then penalizing pseudo-harmful outputs to drive genuinely harmful completions. Additionally, a direction-priority token optimization (DPTO) strategy aligns token updates with gradient directions for faster convergence. Experiments across multiple LLMs show that TAO-Attack surpasses state-of-the-art baselines, achieving consistently higher success rates, even reaching 100% in some cases.

**Strengths:**

- The paper is clearly written, and motivates the proposed approach well in a lucid manner.
- The paper presents detailed evaluations on multiple LLMs.
- The paper propose a novel optimization-based jailbreak method for LLMs that enhances both effectiveness and efficiency, called TAO-Attack.
- Experiments across multiple LLMs show that the TAO-Attack surpasses previous jailbreak methods.

**Weaknesses:**

- Including experiments on more datasets would further strengthen the empirical validation and generalizability of the proposed method.
- Expanding the jailbreak evaluation to additional models—for instance, Qwen series models—could provide deeper insights into the model-specific robustness and transferability of the approach.
- While the current study focuses on jailbreaking LLMs in harmful text generation, it would be valuable to discuss the broader applicability of the proposed techniques to other security-sensitive scenarios, such as information extraction, influence operations, or content-filter evasion across different modalities (e.g., image generation).

**Questions:**

Refer to Weaknesses.

---

> ### Author Response · Authors · 2025-11-21
>
> Thanks very much for your helpful and positive comments.
>
> **1. About more datasets and additional models.**
>
> We evaluate TAO-Attack and I-GCG on the **Qwen2.5-7B-Instruct** model using the HarmBench Behaviors subset. Specifically, we use the same fixed initialization ("! ! ! ! ! ! ! ! ! ! ! ! ! ! ! ! ! ! ! !") for all harmful querys and set a maximum of 1,000 iterations per query. Please refer to **Section 4.5** for more details. For clarity, all results are reported in the format ASR / Iterations.  For comparison, we also include the original AdvBench results in the table below.
>
> | Method     | HarmBench         | AdvBench          |
> | ---------- | ----------------- | ----------------- |
> | I-GCG      | **100%** / 26     | **100%** / 66     |
> | TAO-Attack | **100%** / **11** | **100%** / **21** |
>
> Across both datasets, TAO-Attack remains consistently more efficient. This confirms that our improvements are not specific to AdvBench and generalize reliably across datasets.
>
> **2. About applicability to other security-sensitive scenarios.**
>
> We evaluate three security-sensitive settings: information extraction, influence operations, and multimodal jailbreaking. Since publicly available datasets for the first two tasks are not suitable for jailbreak evaluation, we construct two adversarial prompt sets using GPT-5.1 and manual verification. Please refer to **Appendix A.8** and **Section 4.5** for more details. For multimodal evaluation, we follow MM-SafetyBench and select four harmful samples from each of five categories (Illegal Activity, Hate Speech, Malware Generation, Physical Harm and Fraud). All attacks use the same fixed initialization ("! ! ! ! ! ! ! ! ! ! ! ! ! ! ! ! ! ! ! !") and run for up to 1,000 iterations per harmful query. Information extraction and influence-operation attacks are conducted on **Qwen2.5-7B-Instruct**, and multimodal attacks are conducted on **Qwen2.5-VL-7B-Instruct**. Results are reported as ASR / Iterations.
>
> | Method     | Information extraction | Influence operations | Multimodal Jailbreaking |
> | ---------- | ---------------------- | -------------------- | ----------------------- |
> | I-GCG      | **100%** / 36          | **100%** / 16        | 90% / 248               |
> | TAO-Attack | **100%** / **29**      | **100%** / **9**     | **95%** / **132**       |
>
> Across all three scenarios, TAO-Attack consistently reduces the number of required optimization steps while achieving high attack success rates. These results show that our method generalizes effectively to information extraction, tasks involving harmful manipulation, and multimodal vision–language models.

---

> > ### Comment · Reviewer_e9qu · 2025-11-26
> > **Official Comments by Reviewer e9qu**
> >
> > Thank the authors for the response. I will maintain my score.

---

> > > ### Author Response · Authors · 2025-11-26
> > >
> > > Dear Reviewer e9qu,
> > >
> > > Thank you very much again for your valuable time and effort. We greatly appreciate your insightful and constructive comments.
> > >
> > > Best regards,
> > >
> > > Authors

---

### Author Response · Authors · 2025-11-21
**Summary of Paper Revision**

We thank all reviewers for their helpful and insightful comments, and we respond to each reviewer individually. We now upload a revised version of the paper, with the main updates summarized as follows:

* **Section 4.5**: We add analyses on the switching mechanism, generalization across datasets and models, and computational overhead and efficiency.
* **Appendix A.6**: We expand the evaluation under two adaptive-defense strategies.
* **Appendix A.8**: We add experiments on Information Extraction (IE) and Influence Operations (IO).
* **Appendix A.9**: We add results on different initialization suffixes.
* **Appendix A.10**: We add a hyperparameter study on the refusal set size (K).
* **Appendix A.11**: We provide an extended analysis of DPTO.
* **Appendix A.12**: We add experiments on larger LLMs (13B-scale models).
* **Appendix A.13**: We add an ablation study integrating I-GCG components into TAO-Attack.

---

### Meta-Review · Area_Chair_QpZ7 · 2026-01-03

**Summary:**

This paper presents TAO-Attack, an optimization-based jailbreak attack for LLMs. The method combines a two-stage loss function—designed to suppress refusals and penalize pseudo-harmful outputs—with a Direction-Priority Token Optimization (DPTO) strategy. Empirical results show that TAO-Attach achieves higher success rates and greater efficiency than state-of-the-art baselines.

Reviewers' concerns focused on **experimental completeness**, several reviewers noted the limited scope of evaluation, pointing to the narrow selection of datasets and model scenarios (Reviewers e9qu, yrv3, peix, q9kk). Specific requests included testing performance across more datasets and suffix variations, as well as a direct comparison of computational time—beyond just iteration counts—since different methods incur different per-iteration costs.

In their rebuttal, the authors provided additional experiments, broadening the evaluation to include more datasets and models, and presenting concrete time-cost analyses. These additions largely resolve the experimental limitations initially noted by the reviewers.

**Reviewer Concerns:**

Two issues remain inadequately addressed in the authors' rebuttal.

First, the reviewers’ request to discuss the **broader applicability of the method to other security‑sensitive scenarios**—such as information extraction, influence operations, or cross‑modal content‑filter evasion—falls outside the core focus of this work, which is **text‑generation jailbreaking**. Omitting an extended discussion on these tangential domains does not weaken the integrity or completeness of the paper’s contributions.

Second, while the authors did respond to questions concerning the **theoretical rationale behind the two‑stage loss design** and the **independent contribution of the DPTO strategy**, a more thorough explanation would significantly strengthen the methodological justification and improve the overall persuasiveness of the proposed framework.



## Reviewer Concerns Details

### Reviewer e9qu
#### Addressed Concerns:
- need Experiments on more datasets: Resolved.
- need Evaluation on more models: Resolved.
- need Discussion on broader applicability (e.g., multimodal): *Not* resolved.

### Reviewer yrv3 (All Resolved)
- Need in-depth analysis of why Stage Two works without penalizing successful attacks: Resolved.
- Need evaluation with different initialization suffixes: Resolved.
- Need time cost comparison with other methods: Resolved.
- Need evaluation of the hyperparameter K: Resolved.

## Reviewer YL48 (All Resolved)
- Need automated or semantically guided expansion of the refusal set $R$: Resolved.
- Need justification for not using a unified loss and evaluation of the Rouge-L trigger: Resolved.
- Need clarification on DPTO's independent contribution and generalizability: Resolved.
- Need evaluation under more challenging or adaptive-defense scenarios: Resolved.

## Reviewer q9kk
- Need discussion on computational resource costs: Resolved.
- Need analysis on hyperparameters : Resolved.
- Need evaluation on larger and more complex open-source models (e.g., Llama-2-70B): *Partially* Resolved. The author conducted experiments on the 13B-scale model, but failed to reach the 70B-scale size that the reviewers explicitly mentioned.

## Reviewer peix (All Resolved)
- Need evaluation on more recent models: Resolved.
- Need justification for not directly incorporating both losses: Resolved.
- Need clarification on combination with existing improvement tricks (e.g., I-GCG): Resolved.
- Need correction of writing issues (headers, math symbol, table alignment): Resolved.

**Reviewer Scores:**

### Reviewer e9qu
 - Score remains 6

### Reviewer yrv3
 - Score remains 6

## Reviewer YL48
 - Initial score: 4. The reviewer did not respond to the author.

## Reviewer q9kk
 - Initial score: 6. The reviewer did not respond to the author.

## Reviewer peix
 - Score increased from 4 to 6

---

### Decision · Program_Chairs · 2026-01-26

Accept (Poster)